# Gaussian process inference modelling of dynamic robot control for expressive piano playing

**Luca Scimeca** [ID]◉*, **Cheryn Ng**◉, **Fumiya Iida**

Bio-Inspired Robotics Lab., Dept. of Engineering, Cambridge University Cambridge, United Kingdom

◉ These authors contributed equally to this work.

* ls769@cam.ac.uk

**Data Availability Statement:** All the files are available in following online repository: https://bitbucket.org/lucascimeca/robo_piano_learning/src/master/data/.

**Funding:** This work was funded by the UK Agriculture and Horticulture Development Board

## Abstract

Piano is a complex instrument, which humans learn to play after many years of practice. This paper investigates the complex dynamics of the embodied interactions between a human and piano, in order to gain insights into the nature of humans' physical dexterity and adaptability. In this context, the dynamic interactions become particularly crucial for delicate expressions, often present in advanced music pieces, which is the main focus of this paper. This paper hypothesises that the relationship between motor control for key-pressing and the generated sound is a manifold problem, with high-degrees of non-linearity in nature. We employ a minimalistic experimental platform based on a robotic arm equipped with a single elastic finger in order to systematically investigate the motor control and resulting outcome of piano sounds. The robot was programmed to run 3125 key-presses on a physical digital piano with varied control parameters. The obtained data was applied to a Gaussian Process (GP) inference modelling method, to train a network in terms of 10 playing styles, corresponding to different expressions generated by a Musical Instrument Digital Interface (MIDI). By analysing the robot control parameters and the output sounds, the relationship was confirmed to be highly nonlinear, especially when the rich expressions (such as a broad range of sound dynamics) were necessary. Furthermore this relationship was difficult and time consuming to learn with linear regression models, compared to the developed GP-based approach. The performance of the robot controller was also compared to that of an experienced human player. The analysis shows that the robot is able to generate sounds closer to humans' in some expressions, but requires additional investigations for others.

## 1 Introduction

Since the dawn of robotics there has been an interest in making machines perform artistic and creative tasks in a human-like manner [1]. Music instrument playing, in particular, is an important challenge, because the skills necessary to play music from physical instruments are

(CP 172), and Physical Sciences Research Council (EPSRC) MOTION grant [EP/T00519X/1]. The funders had no role in study design, data collection and analysis, decision to publish, or preparation of the manuscript.

**Competing interests:** The authors have declared that no competing interests exist.

often beyond state-of-the-art robotics technologies, while the attempt to mimic musicians would give us insights into the artistic nature of humans [2].

Piano, among others, is a complex instrument with rich and complex acoustics, which is difficult to master even for humans after many years of training. The production of rich expressive sounds requires appropriate key-press trajectories with a suitable mechanical apparatus. A key-press event, as performed through a finger by a human musician, can therefore not be seen from the point of view of the finger, or the instrument, in isolation. Rather the action of the finger and the instrument is coupled, where the dynamics of the piano are linked to the bio-mechanics and neuromuscular dynamics of the pianist, and their coupling produces rich and complex acoustic energy radiating from the soundboard [3, 4].

Previous attempts to reproduce piano playing by robots mainly focused on two aspects: the mechanical actuation of the fingers and the algorithms for finger motion planning across keys. A large variety of actuation mechanisms was proposed by using DC motors [5], servomotors [6, 7], pneumatic cylinders [8, 9], and tubular solenoids [10]. These actuation mechanisms were then integrated to various control and planning architectures, such as hard-coded motion paths [6], optimal path planning algorithms [5, 7, 10, 11], and more advanced algorithms including collision avoidance [5, 11].

Although these robotics studies demonstrated impressive accuracy and speed for complex music playing, very little attention has been paid to the understanding of delicate embodied interactions of players and instruments for expressive sound generation. So far [4, 12] have analysed the importance of dynamic interactions for expressive playing, but it is still largely unknown how music expressions can be systematically analysed and understood. Generally speaking, expressive piano playing is a manifold problem involving the dynamics of the instrument, note arrangement in music instructions (sheet musics), and player's action, and we are not able to independently investigate each of these components in isolation as they are mutually related to each other [3].

The problem addressed in this paper is therefore the development of a method to systematically analyse the relationship between these three components, by employing a state-of-the-art digital piano, robot arm platform, and a statistical computational tool based on Gaussian Process (GP) inference. For a systematic analysis and comprehensive understanding of the landscape of this framework, we employed a minimalistic approach where we consider 10 basic playing styles, expressed by a single note, with a finger performing key-presses on a piano instrument. As exemplified in later sections, even with this simplified setup, the systematic understanding of expressive piano playing is nontrivial.

For this challenging problem, this paper argues that the relationship between the motor control of a player and the corresponding expressive auditory output on the piano is intrinsically nonlinear, thus specific treatments are necessary when designing and analysing motor control of piano players. The expressive piano playing is known to be analysable by the MIDI format of music sound representations, in which expressive sounds are related to the velocities of piano key-pressing and interval times between them. Based on this framework, we will extend the analysis to robot control to show the non-linearity of the relationship between expressive and a player's motor control. The identification of this nonlinear nature of piano playing is particularly important in order to understand players' (bio)mechanical dynamics, control, and learning processes. In this context, the mechanical dynamics (impedance) of players' fingers, arms and hands are important. Additionally, linear regression methods may not be flexible enough to cope with the nonlinear dynamics of this system, and other nonlinear control optimisation (learning) processes become instead necessary.

In the past, humans have been shown to learn and make decisions with processes akin to Bayesian inference, above all in tasks involving sensory-motor control [13]. It is in this context

that we propose a fully probabilistic GP-based framework to capture the relationship between the piano music and key-press events that generated it. Advantages of this approach include a mathematically meaningful measure of uncertainty in key-press trajectory prediction. While these implications are valid for both human and robot players, it would be particularly interesting for designers of piano playing robots, because a hard-coded linear mapping of motor control would not be sufficient for human-like playing of piano but an integrative view of morphology and sensory-motor control become more valuable in the context of dexterous manipulation tasks [14–16].

This paper is structured as follows. Section 2 reports the methods in this Paper, including the GP-based Learning framework in Section 2.1, and the robotic experimental set up in Section 2.2. In Section 3 we report the results of this work. Finally, in Section 4 a discussion and a conclusion are provided.

## 2 Materials and methods

### 2.1 Learning framework

The framework developed for this work aims to capture the relationship between piano key-press events and the corresponding piano sound outputs, thus optimizing the robot's key-press trajectories for different styles, through a single demonstration. Much like a human player, the robot can perform key-presses on a piano, observe the resulting music output, and then explore its own action space and the consequences of its actions through sound feedback (Fig 1). The music styles chosen for the experiments are commonly used in piano playing to evoke different musical expressions. Two types of fundamental musical parameters governing musical events are explored, articulation and dynamics, for which a musical event is typically a single note or phrase of notes. Music articulations shape the attack, decay and length of an event, while dynamics determine the loudness of an event relative to the entire passage. Articulation methods *tenuto*, *staccatissimo* and *staccato* were chosen for the experiment for their relevance to the piano instrument and suitability for monophonic (single-note) playing. The styles of *fortissimo/ ff* (very loudly) to *pianississississimo/pppp* (extremely softly) were instead chosen for their dynamic range. These playing styles introduce a wide range of music which require very diverse types of key-press action to be performed on the instrument.

**2.1.1 Linear models.**   The first set of models are Linear models, capturing the relationship between the detected key-press piano sounds and their corresponding robot key-press actions. Fig 1a and 1b show a qualitative diagram of the framework in the context of these experiments. For each key-press let $\vec{v}$ be the $d_v$ dimensional vector of control parameters utilized for the robot control, and $\vec{o}$ the $d_o$ dimensional vector of the corresponding sound outputs. An indetail explanation of the parameterization of the robot control and the sound output is irrelevant to the learning framework, and will be provided in later sections.

For notation's sake we will impose $x = \vec{o}$ and $y = \vec{v}_i$. In this context, $\vec{v}_i$ is a one dimensional vector, corresponding to one dimension of the control action parameters (Fig 1b). The following equations will be repeated for every control action dimension, thus $i \in [1 \ldots d_v]$, where $d_v = 5$ for the duration of the experiments.

For the Linear models we impose $x' = [x\ 1]$ and:

$$y = \vec{w}x' \tag{1}$$

where $\vec{w}$ is a $d_o + 1$ dimensional vector of weights capturing the relationship between the sound outputs and the control parameter $\vec{v}_i$ under consideration. The values for $\vec{w}$ are approximated by a Least Square fit.

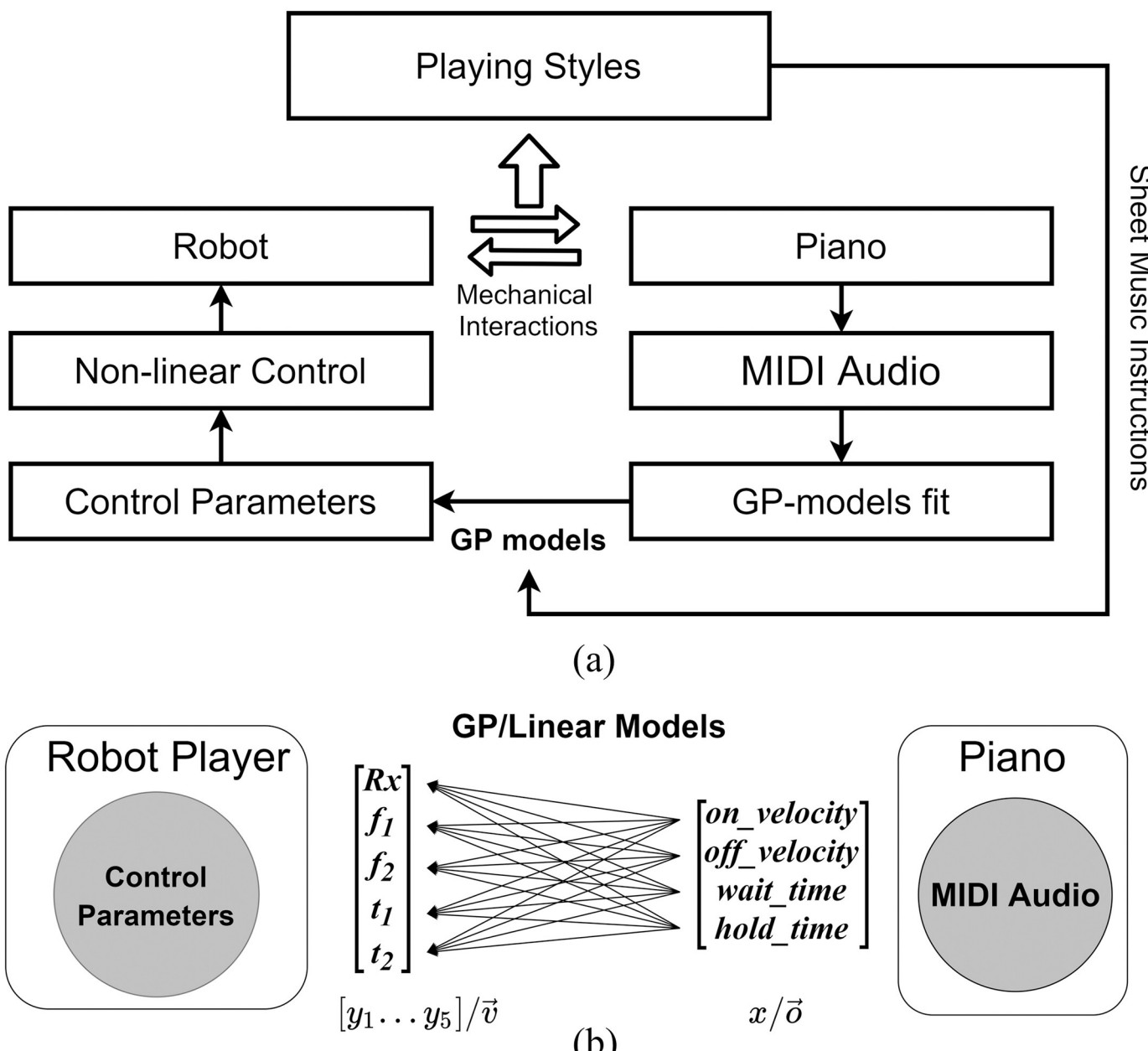

**Fig 1. Adaptive piano playing diagram, including (a) the GP-based framework developed and (b) the model relationship between robot key-press and piano sound outputs.**

**2.1.2 Gaussian process framework.** In the past, humans have been shown to learn and make decisions with processes akin to Bayesian inference, above all in tasks involving sensory-motor control [13]. In contrast to the linear models, a GP-based framework is used to capture the relationship between the sound produced by the piano and the robot control of the key-press generating it [17]. Given key sound observations $x$, generated by noisy robot-controlled key-presses $y$, the relationship of sound output to motor control can be captured by:

$$y = f(x) + \epsilon \quad \text{where } \epsilon \sim \mathcal{N}(0, \sigma_y^2) \tag{2}$$

i.e. the noisy key-press by the robot control are assumed to have a Gaussian process prior and be drawn from:

$$y = f(x) \sim GP(m(x), k(x, x') + \delta_{pq}\sigma_y^2) \tag{3}$$

where the mean is $m(x) = 0$, and the covariance of any two noisy observations $y_p$ and $y_q$ is:

$$cov[y_p, y_q] = k(x_p, x_q) + \delta_{pq}\sigma_y^2 \tag{4}$$

where $x_p$ and $x_q$ are the inputs to the corresponding observations, and $\delta_{pq} = I(p = q)$. The relationship between $y$ and $x$ in Eq 3 is thus dictated by how any two musical outputs co-vary in terms of their generating key-press trajectory. The covariance of any two points is governed by Eq 4, and thus the choice of the kernel is here important. We build on a linear kernel, and account for non-linearities in the relationship of $x$ and $y$ by a Radial Basis Function Kernel, thus:

$$k(x_p, x_q) \quad = x_p x_q \sigma_f^2 \; e^{(-\frac{\|x_p - x_q\|^2}{2l^2})} \tag{5}$$

where $\sigma_f^2$ and $l$ are hyperparameters which decide the magnitude of influence of adjacency when evaluating the function at any one point. From [18], we can write the mean $\mu_*$ and variance $\Sigma_*$ for any new test audio input $X_*$, prior inputs $X$ and generating observed control key-press $y$ as:

$$\mu_* = k(X, X_*)^T K_y^{-1} y \tag{6}$$

$$\Sigma_* = k(X_*, X_*) - k(X, X_*)^T K_y^{-1} k(X, X_*) \tag{7}$$

where $K_y = k(X, X) + \sigma_y^2 I_N$, to account for the noisiness of the observations.

Finally, it is desirable not to manually pick the hyperparameter $\sigma_f^2$ and $l$ of the covariance function. We therefore perform model selection by initializing the $\sigma_f^2$ and $l$ to 1 and iteratively minimizing the negative marginal log likelihood $-\log p(y|X)$ over 100 training steps as implemented in [19].

The equations described can capture the relationship between any sound output parameter $\vec{v}_i$ and control input $\vec{o}$. In this paper $d_v = 5$ and $d_o = 4$, thus five 4-dimensional Gaussian Processes are built to automatically capture the relationship between the sound output and robot key-press control.

## 2.2 Experimental set-up

For the experiments we use a UR5 robotic arm, equipped with a custom end-effector (Fig 2a and 2b). The music instrument is a Kawai Es8 Digital Piano, which provide the possibility to retrieve event-based, MIDI audio messages when a key is pressed. An audio message is generated when one of two events is detected: a key press or a key release. For every pair of detected MIDI message, four variables are going to be relevant for the purpose of the experiments in this paper, namely: the velocity of the key-press, the time the key was held down, the velocity of the key release and the wait time before performing the next key-press; the four variables will be referred to as *on_velocity*, *hold_time*, *off_velocity* and *wait_time* for the remainder of the experiments.

**2.2.1 Finger design.** The finger was designed to be the simplest end-effector to allow the UR5 robotic arm to perform single key-presses on a standard piano. The finger is a $80mm \times 15mm$ cylindrical attachment, with a flat origin and a rounded finish, to perform key-presses

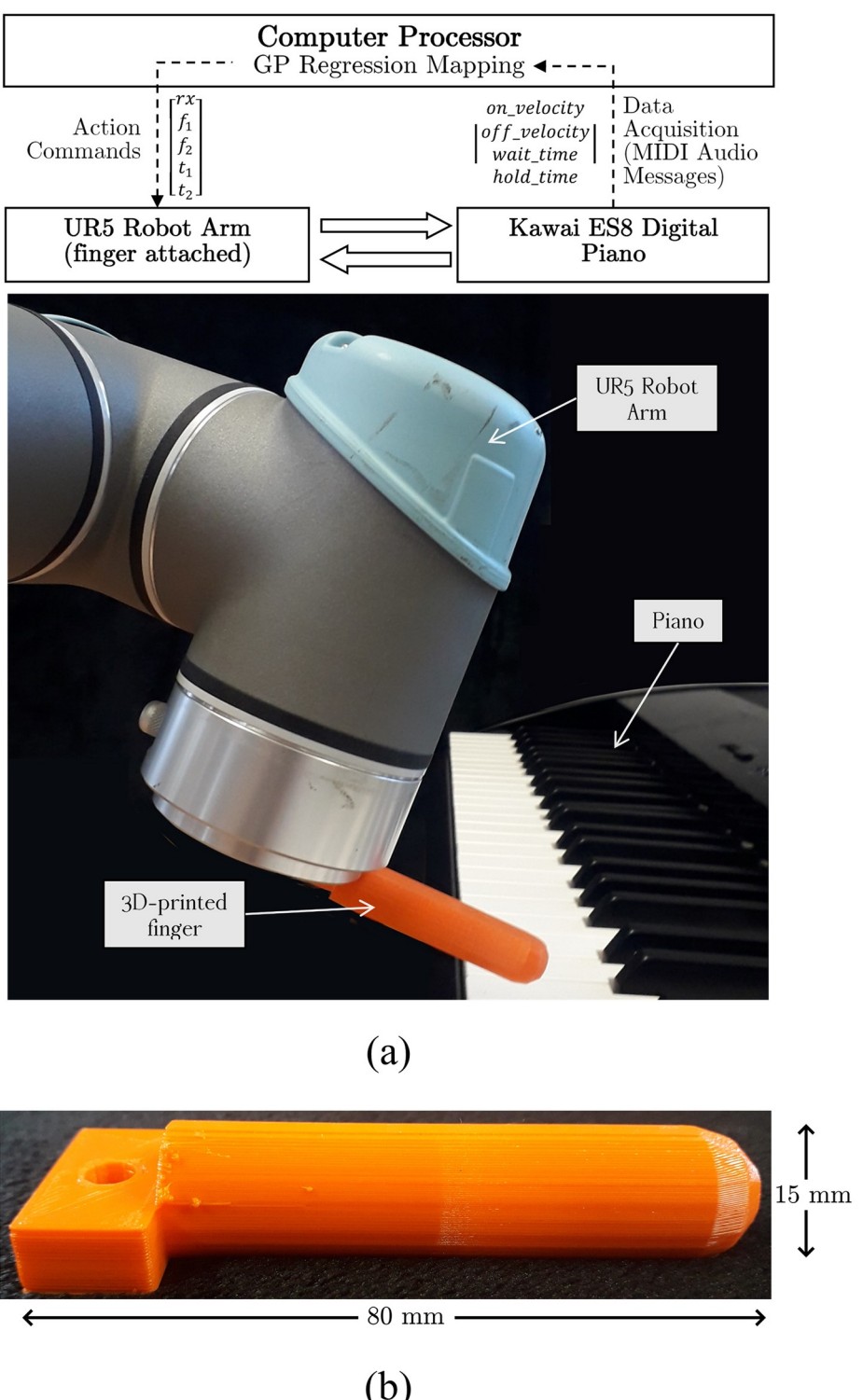

(a)

(b)

**Fig 2. The set-up for the experiments.** Fig. (a) shows robotics set up, including a schematic of the robot connection to a processing unit and the musical instrument. Fig. (b) shows the 3D finger used for piano playing.

at various stroke angles without compromising the area of contact (Fig 2b). The finger was 3D-printed using FilaFlex, a Thermoplastic Polyether-Polyurethane elastomer (TPE) filament of shore hardness 82A https://recreus.com/en/12-filaflex-original-82a, and thus it presents some room from flexing and bending.

## 2.3 Robot control

The robot was controlled in Cartesian coordinates at $\approx 125 Hz$, acting upon the Z and Rx tool axis, to generate the desired contact between the end-effector and piano key for a key-press. The $x$ axis and $Rx$ axis were controlled first, to align the robot's end-effector with the key to be pressed ($x$), and assume a rotation about the fingertip ($Rx$). Subsequently the robot was controlled via the $z$ axis, to perform a key-press. A hybrid sinusoidal displacement profile was generated for the $z$ axis, parameterized in both amplitude and frequency. The alignment, controlled by the $x$ axis, does not generally influence a key-press. The $Rx$ and $z$ axis, instead, influence each key-press uniquely. A total of 5 parameters were used to these two axis during each key-press experiment ($d_v = 5$), i.e. $Rx, f_1, f_2, t_1, t_2$ (Fig 3b).

For the Z axis of motion, a sinusoidal displacement over the course of the key-press is defined as:

$$s_z(t) = \begin{cases} 0 & \text{if } 0 < t \leq T_a \\ \frac{1}{2}A_z\left[\cos\left(\frac{2\pi f_1}{2}(t - T_a)\right) - 1\right] & \text{if } T_a < t \leq T_b \\ -A_z & \text{if } T_b < t \leq T_c \\ -\frac{1}{2}A_z\left[\cos\left(\frac{2\pi f_2}{2}(t - T_c)\right) + 1\right] & \text{if } T_c < t \leq T_d \end{cases} \tag{8}$$

where

$$T_a = t_1 , \ T_b = t_1 + \frac{1}{f_1},$$

$$T_c = t_1 + \frac{1}{f_1} + t_2 , \ T_d = t_1 + \frac{1}{f_1} + t_2 + \frac{1}{f_2}$$

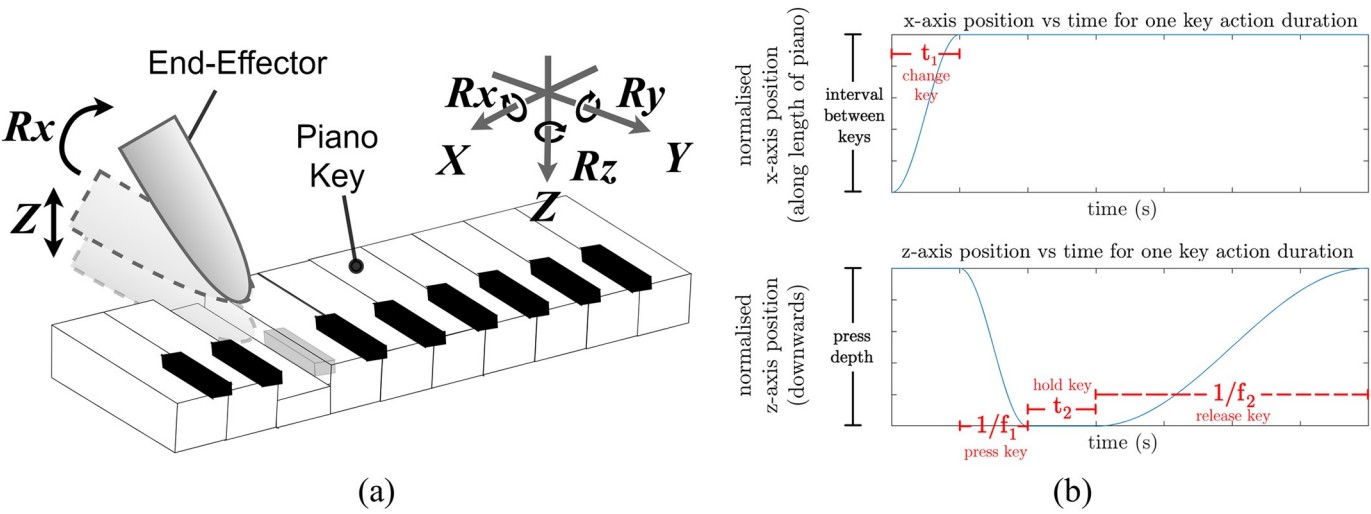

**Fig 3. The robot control, including (a) an illustration of a robotic key-stroke, and (b) the hybrid sinusoidal control designed for the experiments.**

Here, 't' is the time, in seconds, elapsed since the start of the touch experiment and $A_z$ defines the amplitude of the generated sinusoidal displacement for the key press, and it is here set to $32mm$ throughout the experiments. Additionally, a parameter $Rx$ sets the angle of rotation of the end-effector, between 0° and 90° throughout the experiments (Fig 3a). The $x$ and $Rx$ axis of motion controls the robot's ability to shift key along the piano as it plays different notes. The control on the $x$ axis is achieved through:

$$s_i(t) = \begin{cases} \frac{1}{2}A_i[\cos(\frac{2\pi}{2t_1}t) - 1] & \text{if } 0 < t \leq T_a \\ A_i & \text{if } T_a < t \leq T_d \end{cases} \quad (9)$$

$$i \in \{Rx, x\}$$

where $A_x = k_d n_k$, $k_d$ represents the key-width and $n_k$ is the number of keys between the previous and current key. The standard modern piano keyboards are designed with white keys $23.5mm$ wide, thus we set $k_d = 23.6$, taking into account the gap between keys. The following sections will explore how different control parameters can approximate different playing styles, and how these may be learned online through sound feedback.

## 3 Results

In the following sections we wish to understand the delicate embodied interactions of players and instruments for expressive sound generation. We first show how expressive piano playing is a manifold problem, involving the dynamics of the musical instrument, note arrangement, and player's action. Here, we show that the relationship between motor control and piano is intrinsically non-linear. We will further show the viability of the GP-based framework developed in capturing the non-linear dynamic relationship of this system, and its advantages with respect to simpler linear regression methods. Finally, the optimized controllers for 10 different playing styles are compared with the performance of an expert human player.

### 3.1 Robot key-press control to sound feedback

In the first set of experiments we investigate the relationship between the robot control parameters and the generated sound outputs following the robot key-press control. This analysis is based on observations on a large-scale set of over 3125 key-press experiments performed with the set-up described in the previous section. Fig 4 shows example relationships between the MIDI parameters of *on_velocity* and *off_velocity* with the control parameters of $f_1$, $f_2$, and $Rx$. From Fig 4a, it is possible to see how the *on_velocity* increases as the robot control's $f_1$ parameter increases, while all other control parameters are kept constant. However, the normalized value of *on_velocity* saturates at ≈0.4 regardless of an increase in $f_1$ from 6.6 Hz. Beyond a frequency threshold of 6.6 Hz, any higher imposed frequencies in the key-press control appear indistinguishable by the piano key's velocity-sensitive trigger sensor. The piano key trigger has not reached its velocity sensing saturation, as we observe from Fig 4e that at other finger rotation angles ($Rx$), the normalized *on_velocity* values are able to reach up to ≈0.8 compared to the saturation at ≈0.4 previously observed at $f_1 = 6.60$ Hz. It is likely that this is in part due to the elasticity of the finger, which is capable of flexing and bending to some degree, combined with the sinusoidal parameterization of key-press trajectory. Both the finger's make and the choice of action parameterization for the robot key-press control, in fact, induce slight changes in both the stiffness and contact point of the finger with respect to the stroked key during a key-press, with higher angles inducing higher degrees of stiffness in the end-effector.

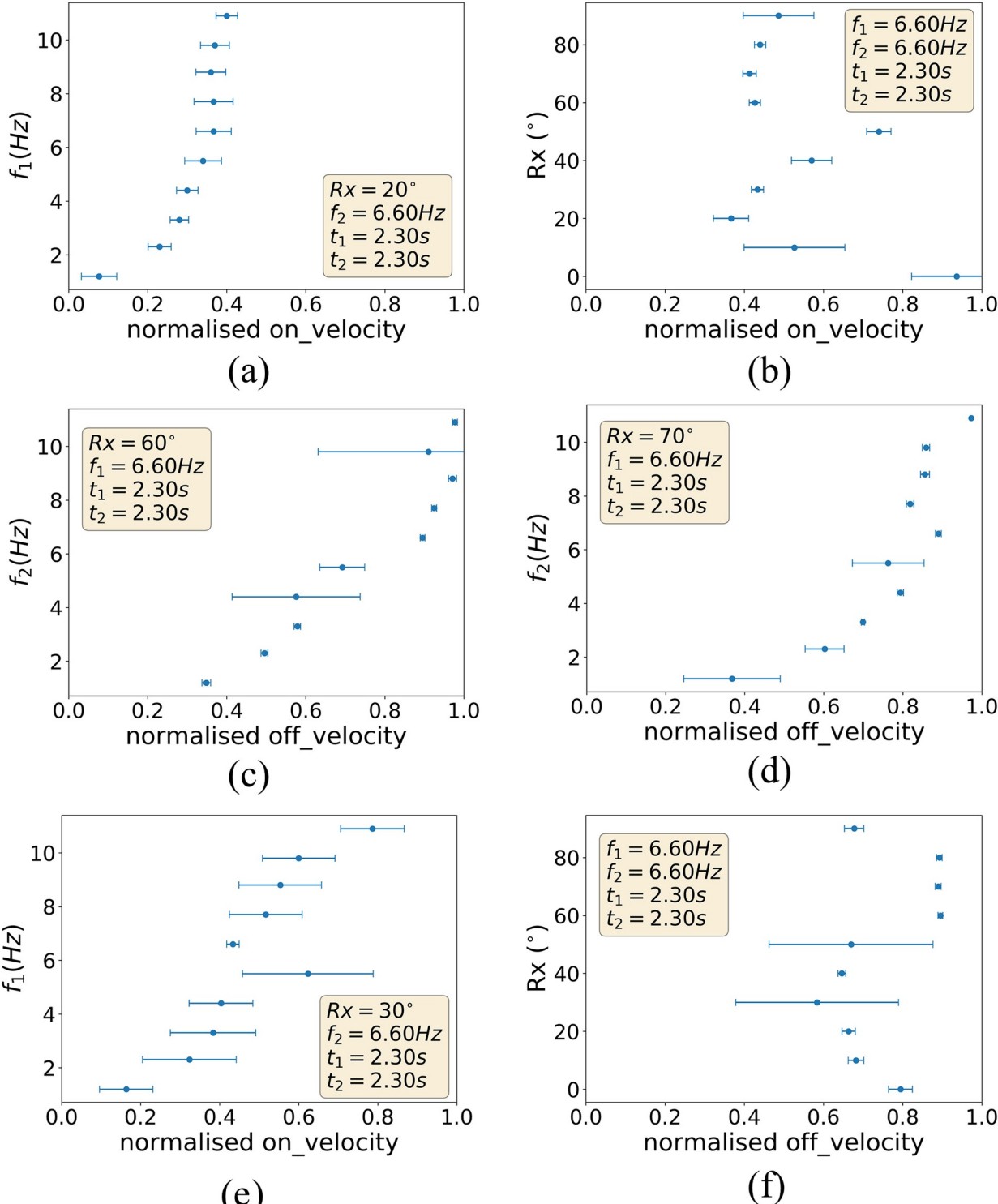

**Fig 4. The sample raw data corresponding to the MIDI sounds registered by the piano and the control parameters generating the key-presses execution, averaged over 10 trials.**

We observe a similar trend between $f_2$ and *off_velocity* in Fig 4c and 4d. The normalized *off_velocity* detected by the piano increases as the $f_2$ parameter is increased in the robot control, while all other control parameters are kept constant. However, the value of *off_velocity* saturates at values of $f_2$ frequencies of $\approx 6.6Hz$. Beyond a frequency threshold of $6.6Hz$, higher release velocities are indistinguishable by the trigger.

The non-linear relationships between *Rx* and audio parameters *on_velocity* and *off_velocity* respectively are harder to capture as there are additional factors at play. As the finger's material is non-stiff material, and given its elongated structural composition, key-presses at different angles may vary the finger' stiffness, as the generated forces derived from the key-press may be more or less normal to its longest side.

The non-linear relationships observed are also not representative of those observed at other constant variable values. This is illustrated by comparing Fig 4a and 4b for which all constant robot control variables are the same except for *Rx* = 20˚ for the data in Fig 4a and *Rx* = 30˚ for that in Fig 4b: there is a less significant plateau observation of *on_velocity* in the latter as the value continues to increase gradually to 0.8 at $f_1$ = 10.9 Hz. This is likely because at the higher rotation angle, the depth at which the piano key is electronically triggered corresponds to a different point along the gradient of the sinusoidal curve in Fig 3b, causing a velocity difference that is distinguishable by the trigger as $f_1$ increases. Other factors may also contribute to this difference, such as the different mechanical properties of the finger at different rotation angles and the differing point of contact of the finger on the key.

Similarly when comparing Fig 4c and 4d, which plot the data obtained from setting *Rx* = 60˚ and 70˚ respectively, we observe a more significant plateau between $f_2$ = 7.7 Hz and $f_2$ = 9.8 Hz in the latter figure, where the velocity change due to $f_2$ is indistinguishable by the key trigger.

The analysis of the raw data shows the complexity, multi-dimensionality and non-linearity of problem at hand, where the physical interaction of the robot's finger and the piano instrument is quantified experimentally.

## 3.2 Gaussian process based framework analysis

As shown in the previous section, the relationship between the control parameters and resulting note musical outputs is both non-linear and multivariate dependent. Gaussian Processes can capture both the non-linear nature of the relationship between the inputs and outputs, and the dependence across parameters.

In the second set of experiments it is shown how the GP-based framework developed can approximate a parametric fit during training. We initially thus ignore the complexity of multivariate fits and run the framework by optimizing a single control parameter with respect to one MIDI output. We chose a control parameter and MIDI output which should show some degree of correlation, e.g. $f_1$ and *on_velocity*, and run the algorithm to train the robot over 12 key-press, or iterations. Fig 5 shows the algorithm at 5 different stages within the 12 iterations. As shown in the figure, for each parametric value attempted by the robot, the uncertainty of the fit at that point collapses, and is later related to the variance of the fit at that point. By iteration 12, the robot has found a fit over almost the whole controllable parameter space. At this point, the GP model trained on the same samples can be used to inference the control necessary to reproduce a wanted MIDI output.

In the next set of analysis we now consider all parameters, i.e. five 4-dimensional GP models are fit, to capture the relationship between the 4-dimensional MIDI audio piano outputs and each of the 5 control parameters. At each iteration, we use Eq 6 on the five 4-dimensional GP models, to choose controls to approximate each of the playing styles shown in Fig 6a. We

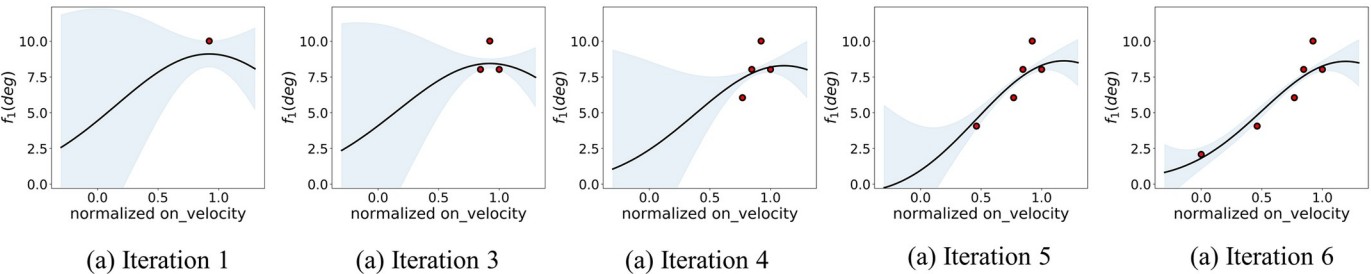

(a) Iteration 1    (a) Iteration 3    (a) Iteration 4    (a) Iteration 5    (a) Iteration 6

**Fig 5. The GP-based exploration fit over different iteration steps when running the framework with simple *f*1 control on the *on_velocity* parameter.**

use MuseScore https://musescore.org/en, a digital score-writer computer program that supports MIDI output, to generate each playing style, and the resulting sound output. The actual sound outputs generated by the inferred control can then be compared to the MIDI outputs to reproduce, and an error can be computed by:

$$error_s = \sqrt{(\vec{o}_{inv,*} - \vec{o}_{s,midi})^T (\vec{o}_{inv,*} - \vec{o}_{s,midi})} \qquad (10)$$

where $\vec{o}_{inv,*}$ is the MIDI output generated by applying the inferred control $\mu_{inv,*}$ of the inverse model, and $\vec{o}_{s,midi}$ is the reference MIDI output for the playing style under consideration.

We compare the robot note error over the 10 different playing styles when learning through the GP-based framework developed, against a linear fit of the $x-y$ parameters in Section 2. In both cases the robot searches the space of each control parameter in a breadth-first grid-search fashion, with a parametric discretization of each parameter into 5 equally spaced values. The robot searches each parameter combinatorially, so a total 3125 key-press are performed to incrementally train the linear and GP models. Fig 7a shows the sound errors of testing key-presses on the piano, after testing the fits every 30 training key-press iterations. For each testing epoch, the robot is made to test each playing style 3 times for the linear model, and three for GP-based framework, bringing the total number of experiments to 9375 piano key-press for both training and testing, with a split of 50% and 50% respectively. From Fig 7a it is clear how the GP-based framework developed is capable of outperforming the simpler linear model,

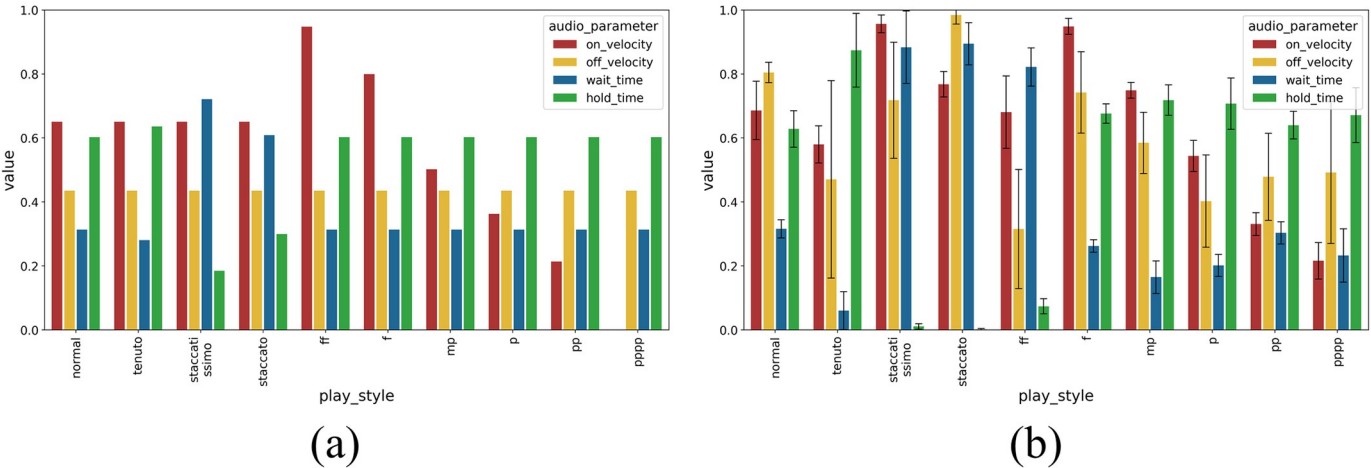

(a)    (b)

**Fig 6. The 10 different playing styles addressed in this work.** (a) The playing styles generated by MuseScore digital score writer, and (b) The play styles as played by the human player. The variance between the MIDI parameters shows the fundamental differences between the various styles.

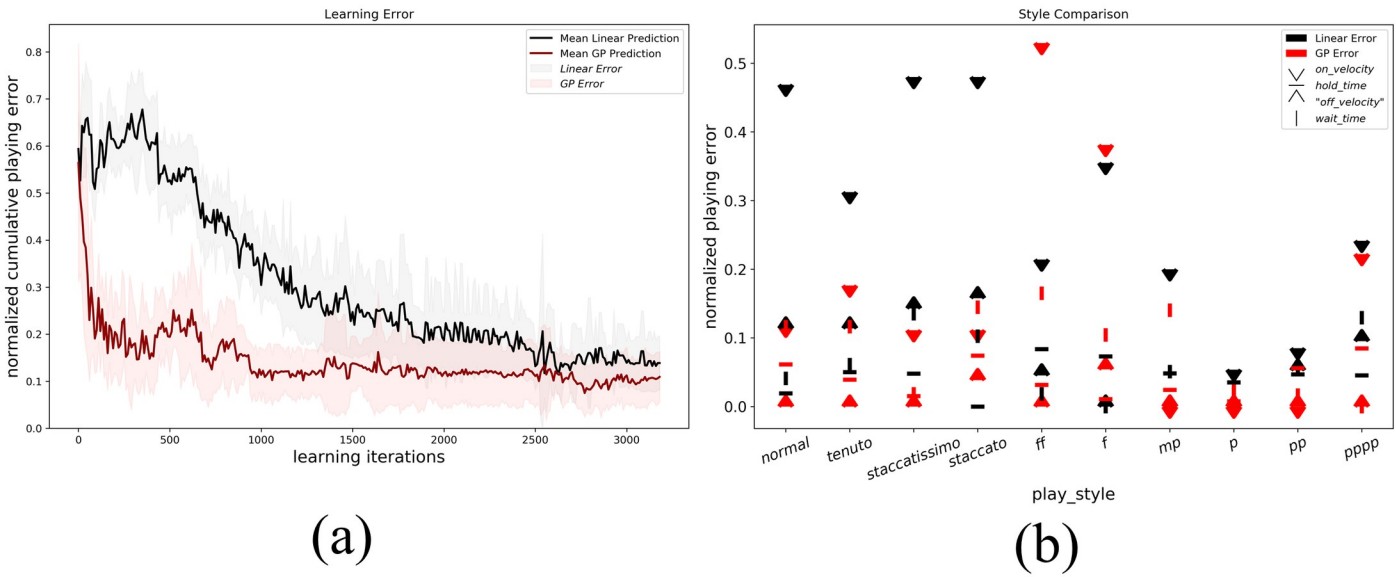

**Fig 7. The comparison between the Linear and GP fits to approximate 10 different piano playing styles.** (a) The testing Error over 3125 training key-press, and (b) the average testing error by play-style, for the best validating epoch during training for the linear and GP models respectively.

bringing the lowest error to 0.0747 MIDI units as opposed to 0.117. More interestingly, the GP-based framework reaches convergence after approximately 1000 iterations, a factor of three times smaller than the time necessary to approximate the playing styles by the linear models.

Fig 7e sheds some light into the limitation of robotic piano playing with a set up analogous to our own. The robot, in fact, is incapable of matching the key-press velocity necessary to approximate each playing styles, when learning off sound feedback through simple linear models. The GP-based framework outperforms the linear models by a larger margin in 8 out of 10 playing styles. The framework results better suited in capturing duration relationships between target temporal patterns and control key-stroke dynamics, additionally to highly accurate control on slow-speed downward key-strokes, effectively reaching lower MIDI errors for the styles of *normal*, *tenuto*, *staccatissimo*, *staccato*, *mp*, *p*, *pp* and *pppp*. The Linear model results capable of better capturing the relationship between high speed robot control for downward key-press actions, regulated by $f_1$, and target louder sound outputs, finally achieving better performance in the styles of *ff* and *f*. The difference in performance for the styles of *f* and *ff* can be explained by the intrinsic tendency of the linear model to overestimate the levels of downward velocities required to achieve high *on_velocity* outputs. The downward velocity levels, regulated by $f_1$, can in fact be observed to saturate at certain levels in Fig 4a and 4b, levels which depend mainly on the finger angle to the piano key. The linear model will not be able to capture the *on_velocity* plateau, overestimating downward key-stroke velocities, but effectively achieving louder outputs for the *f* and *ff* styles.

Fig 8 shows the control parameter values attempted by the robot as generated from the GP model prediction. It is clearly shown that for playing styles *normal*, *tenuto*, *staccatissimo* and *staccato*, control parameters $Rx$, $f_1$ and $f_2$ have very similar values, at approximately 60˚, 8.5$Hz$ and 4.5$Hz$ respectively. The control parameters $t_1$ and $t_2$ vastly vary across these playing styles, showing variations within 0.5$s$, and indicating a large contribution of these in the playing style's unique characteristics. On the other hand, for playing styles *ff*, *f*, *mp*, *p*, *pp*, *pppp*, the change in dynamics, which clearly defines these playing styles' unique characteristics, is largely

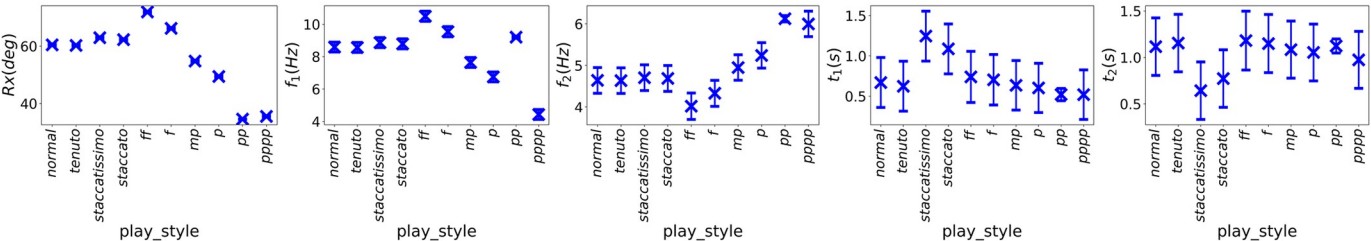

**Fig 8. The final predictions, after 3125 learning epochs, of the required control parameters for 10 different playing styles.** The Confidence intervals are shown as normalized predictive variance, and show the predictive uncertainty of the GP model.

contributed by control parameters $Rx$, $f_1$ and $f_2$, with observed changes of the magnitude of 15˚, 6$Hz$ and 3$Hz$ respectively. Moreover, an invaluable advantage of the GP framework proposed is the uncertainty estimation. Analogously to the prediction computation through Eq 6, we use Eq 7 to compute the uncertainty, or variance, of a control when attempting to generate a target sound output. In Fig 8, the uncertainties are shown in terms of normalized variance for each prediction, to better visualize the plot trends. From the figure it is clear how the robot results more confident in generating both rotations and downward velocities. Temporal parameters (controlled by $t_1$ and $t_2$) and upward velocities (controlled by $f_2$) result somewhat harder to grasp over the different playing styles. The high uncertainty over $t_1$, $t_2$ is indicative of one of two factors: one, that the grid-search parametric exploration of $t_1$ and $t_2$ did not attempt any combination which was close to perform any of the playing styles accurately, and thus no ample evidence is present for the inferred control parameters; two, the robot wait time control through $t_1$ and $t_2$ shows high degrees of variability in terms of the actual wait time outcome. Given further results shown in Section 3.3, the first case is more likely. The $f_2$ parameter is also not capable of achieving full control of piano key-release velocities. This is likely due to the dynamics of the piano key-release action, which limits the speed at which each key springs back to its original position after a key-press. For higher key-lift velocities performed by the robot, and controlled by $f_2$, then, the detected velocity of a key release by the piano will eventually saturate.

## 3.3 Human vs robot piano playing

Finally, we investigate the ability of the robot to perform the 10 different playing styles in Fig 6a as compared to an expert human pianist. We use the controllers optimised by the GP-based framework developed. The human performer is a veteran pianist with 15 years of history in piano playing. To perform this comparison, we do not track the human hand trajectory whilst performing the key-press. Instead, we focus directly on the outcome of this trajectory by recording MIDI events during the human playing, and directly comparing these events with the ones triggered by the robot playing. Upon listening to the sound output the pianist is made to reproduce the note on the piano. We collect 10 different key-press samples at 40 beats per minute (BPM) or 1 key-press every 1.5s, performed 4 times by the human pianist, for each playing style, so as to have an idea of the playing variation within each style. The resulting normalised MIDI output from the pianist's playing is shown in Fig 6b. We let the robot perform according to the $\mu_{inv,*}$ (Eq 6) extracted for each playing style after learning through 3125 iterations, log the resulting $\vec{o}_{inv,*}$ from the robot playing and $\vec{o}_{human,*}$ from the human player, and compute errors with respect to the computer generated MIDI for each playing styles.

Fig 9 compares the human and robot's normalized error performance for each playing style. Fig 9a shows the error of the human and robot's performances in terms of *on_velocity*.

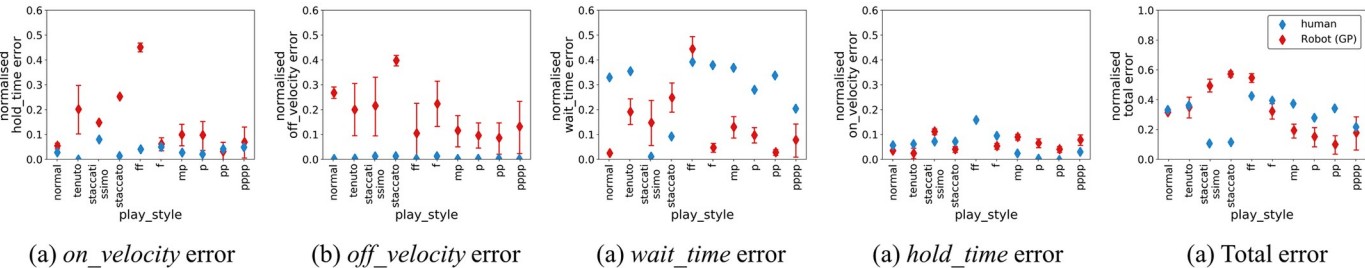

(a) *on_velocity* error (b) *off_velocity* error (a) *wait_time* error (a) *hold_time* error (a) Total error

**Fig 9. Comparison of the playing score of the robot optimized to play the 10 different playing styles after 3125 learning epochs, a human player playing the same, and the computer generated outputs.**

The performances of the robot and the human are highly similar for most playing styles, on average within 0.05 normalized error units from each other for all playing styles. The GP-optimized robot's errors are 0.05 and 0.03 units higher than the human player at *ff* and *f* as the robot is unable to achieve equally high downward velocity required for the loud playing. On the other hand, the robot's errors are lower at *mp*, *p*, *pp* and *pppp*, with error differences ranging from 0.05 to 0.02 normalized units. The precise motion control at low velocities achieved by the robot is, in fact, capable of precisely approximate soft key-presses, which the robot optimizes with respect to the reference MIDI target style. Fig 9b shows the differences between the human player and the MuseScore generated playing styles in terms of *off_velocity*. Due to an innately more dynamic and highly varied key release motion by the human player, the human tends to show a diverse range of release velocities, with errors of up to 0.4 normalized units. The robot, on the other hand, has low variance and error across all play styles due to its consistent speed control for key releases. In Fig 9c we compare the human and the robot performance over the *wait_time* parameter, i.e. the time necessary to wait between key-presses for each playing style. Surprisingly, the robot playing error results higher than the human of 0.1 to 0.4 normalized units in most playing styles, as it is unable to play a melody consistently at 40 BPM with the required waiting time between notes. This is likely due to the inherent delays in the robot's online control when switching commands between key-presses, a consequence of the chosen parameterization and robot key-press control in the experiments, while the human player has a good grasp of rhythm and plays each note at consistent intervals. The robot performs more consistently than the human player in terms of *hold_time* error, with errors lower than 0.1 normalized units across playing styles as shown in Fig 9d, due to the precise clock control during the holding phase of the key-press. In terms of *wait_time* and *hold_time*, the human player's style errors are higher than the robot's for *tenuto*, *staccatissimo*, *staccato* and *ff*, with error value differences varying from 0.1 to 0.5 normalized units. The timing of these articulation styles are exaggerated by the human for greater impact and variation in expression, thus deviating further from the MuseScore generated ground truth. Also note that the error for *wait_time* and *hold_time* (Fig 9c and 9d) show similar trends for the human player, due to the aforementioned good grasp of rhythm; there is no delay between notes and a longer *wait_time* is always compensated by a shorter *hold_time* for that note played. On the other hand, the robot's delays between key-press commands are strongly reflected only in the *wait_time* error. Finally, the overall normalized error by play-style shown in Fig 9e shows an interesting picture. After 3125 learning epochs, the robot is able to perform similarly to the human player for *normal*, *tenuto* and *pppp* playing styles, with normalized error differences lower than 0.01 units. The robot achieves lower errors ranging from 0.1 to 0.4 normalized units for the syles of *staccatissimo*, *staccato* and *ff*, largely due to its accurate *off_velocity*, *wait_time* and *hold_time* performance. The robot, however, performs with errors between 0.1 and 0.2 normalized units

higher than the human player for the styles of *f*, *mp*, *p*, and *pp*, largely due to its poor *wait_time* performance.

## 4 Discussion and conclusion

We investigate the ability for a robot to play the piano according to 10 different playing styles, like a human player. We propose a GP-based framework for the robot to incrementally model the relationship between the control utilized for piano key-press actions, to the resulting sound output, and learn appropriate controllers to play according to each music style. We show that the relationship between control and sound is non-linear in nature, and that different control parameters are not independent with respect to the generated note from the corresponding key-press. The GP-based model can faithfully capture the relationship between control and generated music output, outperforming simpler linear model.

To be able to play different playing styles faithfully it is necessary for the robot to explore its action space, so to find appropriate key-press for each style. The resulting combinatorial explosion in parametric search presents itself as an issue. A second advantage of the proposed GP-based framework is its ability to quickly converge to appropriate controllers for each style. In fact, we observe the GP convergence to be a factor of 3-times faster than linear models, with respect to the learning of the playing styles considered in this work.

The main limitation of the approach lies with the drawbacks of GP modelling. As the model takes into account every single sample to compute the fit, it can eventually be computationally expensive to fit the control to MIDI relationship. This, can in part be obviated by methods which do not need a full kernel representation, and by the dismissal of points far away in time with a sliding learning windows [20].

Finally we compare the ability of the robot to approximate each of the playing styles, with respect an expert human player. We show the comparison sheds some light to several interesting aspects of robotic piano playing. The robot is capable of performing comparatively to the human player in the syles of *normal*, *tenuto*, *staccato* and *pppp*, largely due to the precise control at low speeds, and clock waiting times. The human-player, however, exhibits a much more dynamic and varied playing, which allows them to achieve lower style error to the Muse-Score generated playing styles in *ff*, *f*, *mp*, and *p*. These styles, in fact, require higher downward key-press speeds and dynamic playing. A limitation factor, in this context, is the nature of the comparison, which was applied directly to the MIDI outcome of a key-press. A future interesting direction involves the employment of technology capable to track the trajectory of motion of the human hand whilst performing a key-press [21, 22]. This trajectory can then be compared to the optimized key-press performed by the robot to gain additional insights as to how machine can approach human capabilities for instrument playing.

The dynamic and varied behaviour exhibited by the human player is one of the many advantages complex tools like human hands can possess. Partly, the lack of dynamism is indeed due to the stiffness and simplicity of the robotic end-effector. With the advent of soft-robotics and continuum robots, however, these limitations can be revoked, and the next generation of robots might indeed be able to move away from stiff and *hard-robotics* solutions, towards a softer human-like touch [23, 24]. These experiments shed some light into the limitations of robotic-piano playing, and the issues to be faced when attempting to go beyond monotonic piano playing.

## Author Contributions

**Conceptualization:** Luca Scimeca, Cheryn Ng, Fumiya Iida.

**Data curation:** Luca Scimeca, Cheryn Ng.

**Formal analysis:** Luca Scimeca.

**Funding acquisition:** Fumiya Iida.

**Investigation:** Luca Scimeca, Cheryn Ng.

**Methodology:** Luca Scimeca, Cheryn Ng, Fumiya Iida.

**Project administration:** Luca Scimeca, Fumiya Iida.

**Resources:** Luca Scimeca, Fumiya Iida.

**Software:** Luca Scimeca, Cheryn Ng.

**Supervision:** Luca Scimeca, Fumiya Iida.

**Validation:** Luca Scimeca, Cheryn Ng.

**Visualization:** Luca Scimeca, Cheryn Ng.

**Writing – original draft:** Luca Scimeca, Cheryn Ng.

**Writing – review & editing:** Luca Scimeca, Fumiya Iida.

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
