## [Decision Letter · Decision Letter 0]

23 Jun 2020

PONE-D-20-12759

Gaussian Process inference modelling of dynamic robot control for expressive piano playing

PLOS ONE

Dear Dr. Scimeca,

Thank you for submitting your manuscript to PLOS ONE. After careful consideration, we feel that this paper can be considered for acceptance after minor revision. We invite you to submit a revised version of the manuscript that addresses the points raised during the review process.

Please submit your revised manuscript in 4 weeks (before July 23). If you will need more time than this to complete your revisions, please reply to this message or contact the journal office at plosone@plos.org. Please include the following items when submitting your revised manuscript:

We look forward to receiving your revised manuscript.

Kind regards,

Li Wen

Academic Editor

PLOS ONE

Additional Editor Comments:

Playing piano is a complex skill, and make robot own this skill is a hot topic in recent years. This paper develop the gaussian process inference modelling of dynamic robot control for expressive piano playing. The work is intersting and the experiments are sufficient to prove the proposed methods. I suggest that this paper can be accepted after the authors addressing the comments of the reviewer.

Journal Requirements:

Reviewers' comments:

Reviewer's Responses to Questions

**Comments to the Author**

1. Is the manuscript technically sound, and do the data support the conclusions?

Reviewer #1: Yes

2. Has the statistical analysis been performed appropriately and rigorously? 

Reviewer #1: Yes

3. Have the authors made all data underlying the findings in their manuscript fully available?

Reviewer #1: Yes

4. Is the manuscript presented in an intelligible fashion and written in standard English?

Reviewer #1: Yes

5. Review Comments to the Author

Reviewer #1: Playing piano is a complex skill, and make robot own this skill is a hot topic in recent years. This paper develop the gaussian process inference modelling of dynamic robot control for expressive piano playing. The work is intersting and the experiments are sufficient to prove the proposed methods. However I have some questions,

(1) the paper "investigates the complex dynamics of the embodied interactions between a human and piano, in order to gain insights into the nature of humans' physical dexterity and adaptability". can you explain how do you investigate? with some date golve like Ref."3D human gesture capturing and recognition by the IMMU-based data glove, Neurocomputing, 2018, 277：198-207." or visual way like ” Vision-based teleoperation of shadow dexterous hand using end-to-end deep neural network, International Conference on Robotics and Automation (ICRA), 2019, 416-422.", please help give the explainsion.

(2) As shown in Fig.3, the movement of x-axis is before the movement of z, why are they moving simultaneously?

(3) Please provide the high-quality figures.

6. PLOS authors have the option to publish the peer review history of their article (what does this mean?). If published, this will include your full peer review and any attached files.

Reviewer #1: No

---

## [Author Response · Author response to Decision Letter 0]

30 Jun 2020

We wish to thank the reviewer for the insightful comments. We believe that by addressing the key points mentioned by the reviewer, the quality of the paper has since improved. Thanks to the reviewer’s comments it is now clearer the context in which the experiments were done, including how the ‘human’ playing was investigated, and how the experiments were carried out. The changes are highlighted in red in the revised text “manuscript_tracked.pdf”. The submission folder has also been revised accordingly, with high resolutions figures.

Reviewer #1: Playing piano is a complex skill, and make robot own this skill is a hot topic in recent years. This paper develop the Gaussian process inference modelling of dynamic robot control for expressive piano playing. The work is interesting and the experiments are sufficient to prove the proposed methods. However I have some questions,

(1) the paper "investigates the complex dynamics of the embodied interactions between a human and piano, in order to gain insights into the nature of humans' physical dexterity and adaptability". can you explain how do you investigate? with some date glove like Ref."3D human gesture capturing and recognition by the IMMU-based data glove, Neurocomputing, 2018, 277：198-207." or visual way like ” Vision-based teleoperation of shadow dexterous hand using end-to-end deep neural network, International Conference on Robotics and Automation (ICRA), 2019, 416-422.", please help give the explanation.

We wish to thank the reviewer for this comment. The paper was indeed not very clear as to how the relationship between the human piano playing and the robot playing was captured. The human playing, as well as the robot playing, were not captured by any tracking system. In terms of piano playing, the work focused instead on the outcome of a certain key-press action, rather than how this was performed by the human. In this context, the human hand was not tracked. Instead, the result of the human key-press action was captured by the piano itself in terms of key-press velocity, key-press hold time, key-press release velocity, and key-press wait time. These parameters were initially generated by the MuseScore software, and later attempted by both the human and the robot for comparison.

Human hand tracking technology is indeed a very exciting future direction for this research, one which is ongoing at the moment, and the authors agree it should be mentioned in the manuscript. 

The manuscript has been revised following the reviewer’s comment. We have clarified the explanation of the human and robot playing style capture in Section 3.3, paragraph 1, clarifying no human hand tracking was used for the experiments. The paragraph now reads:

“Finally, we investigate the ability of the robot to perform the 10 different playing styles in Fig. 6a as compared to an expert human pianist. We use the controllers optimised by the GP-based framework developed. The human performer is a veteran pianist with 15 years of history in piano playing. To perform this comparison, we do not track the human hand trajectory whilst performing the key-press. Instead, we focus directly on the outcome of this trajectory by recording MIDI events during the human playing, and directly comparing these events with the ones triggered by the robot playing. Upon listening to the sound output the pianist is made to reproduce the note on the piano. We collect 10 different key-press samples at 40 beats per minute (BPM) or 1 key-press every 1.5s, performed 4 times by the human pianist, for each playing style, so (…)”

We have also augmented the Discussion and Conclusion (Section 4), paragraph 4, mentioning human hand tracking within the context of piano playing as a future direction for this project, we have included the suggested references and expanded on the benefits of this future direction. The section now reads:

“Finally we compare the ability of the robot to approximate each of the playing styles, with respect to an expert human player. We show the comparison sheds some light to several interesting aspects of robotic piano playing. The robot is capable of performing comparatively to the human player in the syles of normal, tenuto, staccato and pppp, largely due to the precise control at low speeds, and clock waiting times. The human-player, however, exhibits a much more dynamic and varied playing, which allows them to achieve lower style error to the MuseScore generated playing styles in ff, f, mp, and p. These styles, in fact, require higher downward key-press speeds and dynamic playing. A limitation factor, in this context, is the nature of the comparison, which was applied directly to the MIDI outcome of a key-press. A future interesting direction involves the employment of technology capable to track the trajectory of motion of the human hand whilst performing a key-press[21, 22]. This trajectory can then be compared to the optimized key-press performed by the robot to gain additional insights as to how machine can approach human capabilities for instrument playing.”

(2) As shown in Fig.3, the movement of x-axis is before the movement of z, why are they moving simultaneously?

We wish to thank the reviewer for this comment. From Fig.3 it is not very clear how the movement across the x and z axis unfolds through the course of the experiments. In the context of the experiments, the movement on the x and z axis do not happen simultaneously. The motion on the x axis is the first to happen, where the robot alights its end-effector with the piano key to press. Afterwards, repetitive experiments can be triggered, where the z and Rx axis can be controlled for each key-press experiment. For each experiment, the Rx axis is the first to be controlled, where the finger will assume a specific rotation position before performing the key-press. One this rotation position is reached, the robot can move along the z axis to perform the key press, before returning to the starting position, and repeating the experiments. We have revised and improved the explanation in Section 2.3, where Paragraph 1 now reads:

“The robot was controlled in Cartesian coordinates at 125Hz, acting upon the Z and Rx tool axis, to generate the desired contact between the end-effector and piano key for a key-press. The x axis and Rx axis were controlled first, to align the robot's end-effector with the key to be pressed (x), and assume a rotation about the fingertip (Rx). Subsequently the robot was controlled via the z axis, to perform a key-press. A hybrid sinusoidal displacement profile was generated for the z axis, parameterized in both amplitude and frequency. The alignment, controlled by the x-axis, does not generally influence a key-press. The Rx and z axis, instead, influence each key-press uniquely. A total of 5 parameters were thus used to control these two axes during each key-press experiment (dv=5), i.e. Rx, f1, f2, t1, t2 (Fig. 3b).”

While Paragraph 3 now reads:

“Here, ‘t’ is the time, in seconds, elapsed since the start of the touch experiment and Az defines the amplitude of the generated sinusoidal displacement for the key press, and it is here set to 32mm throughout the experiments. Additionally, a parameter Rx sets the angle of rotation of the end-effector, between 0 degrees and 90 degrees throughout the experiments (Fig. 3a). The x and Rx axis of motion controls the robot's ability to shift key along the piano as it plays different notes. The control on the x axis is achieved through (…)”

(3) Please provide the high-quality figures.

Thank you for this comment, we have provided a high quality versions of all the figures together with the revised submission.

---

## [Decision Letter · Decision Letter 1]

4 Aug 2020

Gaussian Process inference modelling of dynamic robot control for expressive piano playing

PONE-D-20-12759R1

Dear Dr. Scimeca,

We’re pleased to inform you that your manuscript has been judged scientifically suitable for publication and will be formally accepted for publication once it meets all outstanding technical requirements.

Kind regards,

Li Wen

Academic Editor

PLOS ONE

Additional Editor Comments (optional):

Reviewers' comments:

Reviewer's Responses to Questions

**Comments to the Author**

1. If the authors have adequately addressed your comments raised in a previous round of review and you feel that this manuscript is now acceptable for publication, you may indicate that here to bypass the “Comments to the Author” section, enter your conflict of interest statement in the “Confidential to Editor” section, and submit your "Accept" recommendation.

Reviewer #1: All comments have been addressed

2. Is the manuscript technically sound, and do the data support the conclusions?

Reviewer #1: Yes

3. Has the statistical analysis been performed appropriately and rigorously? 

Reviewer #1: Yes

4. Have the authors made all data underlying the findings in their manuscript fully available?

Reviewer #1: Yes

5. Is the manuscript presented in an intelligible fashion and written in standard English?

Reviewer #1: Yes

6. Review Comments to the Author

Reviewer #1: It is an interesting work, and the paper has been greatly improved according to the reviews, I think it can be accepted.

7. PLOS authors have the option to publish the peer review history of their article (what does this mean?). If published, this will include your full peer review and any attached files.

Reviewer #1: No

---

## [Editor Report · Acceptance letter]

6 Aug 2020

PONE-D-20-12759R1 

Gaussian Process inference modelling of dynamic robot control for expressive piano playing 

Dear Dr. Scimeca:

I'm pleased to inform you that your manuscript has been deemed suitable for publication in PLOS ONE. Congratulations! Your manuscript is now with our production department. 

Kind regards, 

on behalf of

Prof. Li Wen 

Academic Editor

PLOS ONE